# Current and Near-Future Technologies to Quantify Nanoparticle Therapeutic Loading Efficiency and Surface Coating Efficiency with Targeted Moieties

**DOI:** 10.3390/bioengineering12040362

**Published:** 2025-03-31

**Authors:** Vy Tran, Na Nguyen, Scott Renkes, Kytai T. Nguyen, Tam Nguyen, George Alexandrakis

**Affiliations:** Department of Bioengineering, University of Texas at Arlington, Arlington, TX 76010, USA; vy.tran2@mavs.uta.edu (V.T.); na.nguyen2@mavs.uta.edu (N.N.); sar9179@mavs.uta.edu (S.R.); knguyen@uta.edu (K.T.N.)

**Keywords:** nanoparticle characterization, coating efficiency, active targeting nanoparticles, gene delivery, nanopore, single-molecule detection, nanosensor, payload quantification

## Abstract

Active targeting nanoparticles are a new generation of drug and gene delivery systems with the potential for greatly improved therapeutics delivery compared to conventional nanomedicine approaches. Despite their potential, the translation of active targeting nanoparticles faces challenges in production scale-up and batch consistency. Accurate quality control methods for nanoparticle therapeutic payload and coating characterization are critical for attaining the desired levels of batch repeatability, drug/gene loading efficiency, targeting molecule coating effectiveness, and safety profiles. Current limitations in nanoparticle characterization technologies, such as relying on ensemble-average analysis, pose challenges in assessing the drug/gene content and surface modification heterogeneity, which can greatly affect therapeutic outcomes. Single-molecule analysis technologies have emerged as a promising alternative, offering rich information on heterogeneity and stochastic variations between nanoparticle batches. This review first evaluates and identifies the challenges of traditional nanoparticle characterization tools that rely on indirect, bulk solution quantification methods. Subsequently, newly emerging characterization technologies are introduced for the quantification of therapeutic loading and targeted moiety coating efficiencies with single-nanoparticle resolution, to help guide researchers towards advancing the translation of active targeting nanoparticles into the clinical setting.

## 1. Introduction

Nanoparticles (NPs) are engineered nanosized systems that can capture various therapeutic payloads such as small molecules, proteins, and nucleic acids [1]. The use of NPs for drug formulations can improve the therapeutic efficacy of the drug due to their capability to control the dosage, deliver site-specific targeting, enhance bioavailability, reduce toxicity, minimize side effects, and improve patient compliance [2,3]. Due to their superior delivery performance compared to free drugs, the field of nanomedicine has been continuously explored since the 1960s [3]. Doxil, a liposome loaded with doxorubicin, was the first NP to be approved for the treatment of cancer by the Food and Drug Administration (FDA) in 1995 [3,4]. Since then, research on NPs has expanded greatly with many new classes of NPs that are coated and/or loaded with novel agents to improve the efficiency of nanoparticle-mediated therapies. This review focuses on the current status, challenges, and future prospects of drug/gene delivery NPs, especially the characterization of the efficacies of the payload encapsulation and targeted ligand coating, whereas the other aspects of targeted drug delivery are addressed elsewhere [5].

Active targeting NPs are coated with moieties (peptides, antibodies, aptamers) that bind with high specificity to diseased tissues, which have upregulated expression of the targeted biomarkers [6], and selectively deliver their payload at the diseased sites. As specific targeting can enhance cellular uptake, this strategy has been investigated for a wide range of applications, such as cancer therapy, cardiovascular disorders, and infectious diseases [7,8,9]. Although NP types and their applications continue to expand, this technology remains challenging in its translational aspects. Since 2019, only a few NPs have been officially approved by the FDA or the European Medicines Agency (EMA) [10]. In addition, despite the proposed advantages, no active targeting NPs have moved past clinical trials [11]. Due to their more complex design, the production process of active targeting NPs faces more challenges in scale-up production and batch-to-batch consistency [11]. In order for the products to be more reliable, they require more characterization steps and a longer development timeline, which increases costs [11].

Incorporating genetic materials with NPs for gene therapy is another strategy of scientists to enhance delivery efficiency, reduce toxicity, and increase the cell targeting. Gene therapy has several advantages compared to other therapies since the main goal of this strategy is to address the root cause of the disease. Genetic materials such as RNA and DNA can be loaded inside viral or non-viral vectors to be delivered specifically to the cytosol or nuclei of cells to transfect the cells. Some notable NPs for gene therapy are mRNA lipid nanoparticles (LNPs) for COVID vaccines from Pfizer and Moderna and Patirisan to treat hereditary transthyretin [12,13]. Extracted exosomes, which naturally carry genetic materials, are also a cutting-edge strategy [14]. Despite their novelty, not a lot of nanoparticle types for gene delivery have been approved to date, with the main reasons being challenges with scale-up in production, batch-to-batch consistency, and associated costs. These widely used NPs are passive, and there is a lack of targeted commercial NPs.

The accurate characterization of nanoparticle formulations can help determine the repeatability and process efficiency, improve therapeutic activity, and ensure the safety of the product [15]. Better characterization of NPs can lead to more homogenous NP preparations to reduce the risks of dose-dependent cytotoxicity and genotoxicity that have been observed *in vitro* and *in vivo* [16,17]. However, current characterization methods for NPs rely on ensemble-average (bulk analysis) studies, which only report average measurements over large numbers of nanoparticles. For example, the commonly used dynamic light scattering (DLS) method is not efficient at analyzing complex, polydisperse nanoparticle suspension [15]. DLS can create bias because scattered light from large-size particles tends to dominate over the signal of smaller particles in the sample [15].

Current limitations of targeted nanoparticle characterization are the inconsistency in the drug content loading and surface modification uniformity during production due to a lack of standardized physicochemical characterization procedures [18]. A case study showed that batch-to-batch inconsistency of PEGylation coating on gold nanoparticles resulted in different results in vivo [19]. Additionally, a comparison study showed that different traditional techniques that have been used to record the average amount of drug content reported inconsistent data [20]. Furthermore, quality control methods for nanoparticle coatings with targeted ligands or nanoparticles with payloads have many weaknesses. Currently, we do not know how many ligands are, or can be, attached per nanoparticle, nor do we know the ratio between coated and uncoated nanoparticles. Additionally, we currently do not have any characterization tools to quantify the payload for a single nanoparticle, and direct methods for quantifying the payload are still lacking. Clearly, there is a lack of adequate quality control at the present time to maximize the therapeutic efficiency of drug delivery systems and to minimize drug side effects while also reducing the wastefulness of drugs and targeting ligands used and reducing production costs. Therefore, there is a need to improve quality control for targeted nanoparticles and to help push the translation progress of nanotechnology into the clinic [21].

A recent positive development is the invention of single-molecule and single-particle characterization assays that can provide far richer information compared to their ensemble-averaged counterparts. For example, Refeyn’s mass photometry or Nanolyze devices are powerful methods for characterizing a variety of analytes, including proteins, nucleic acids, viruses, and the initial drug delivery systems such as vesicles, micelles, and adeno-associated (AAV) vectors [22,23,24,25,26,27,28]. This label-free technology provides detailed information on the true molecular mass and molecular heterogeneity of the analytes, which may be obscured when using ensemble methods such as DLS. Single-molecule analysis can provide some information that ensemble-average analysis cannot provide such as heterogeneity and stochastic variability [29] and complex chemical kinetics [30,31]. One of the benefits of single-molecule analysis is that it can detect rare events that can be lost on ensemble-average analysis [29]. The rare events can be nanoparticles that are too large to be utilized or nanoparticles that do not have any coating or loading. The advantage of having a full distribution rather than just having a mean value measured is that it can reveal the quality of each batch and help improve the scalability, which can potentially help to transition to the clinic better. Therefore, developing new techniques for single-molecule analysis can provide a complementary system to help overcome the limitations of traditional ensemble techniques.

In this work, we review both traditional and state-of-the-art methods for characterizing nanoparticles, including quantifying drug contents and surface modification, emphasizing the need for more precise and accurate measurements such as single-molecule characterization. Additionally, we aim to identify the challenges associated with these methods and propose potential solutions for facilitating nanoparticle transitioning into the clinical setting. Lastly, we hope that this review will provide insights that assist scientists in selecting the most suitable characterization technologies for their needs. 

## 2. Nanoparticles for Active Targeting Delivery

Most nanoparticles injected into the body are expected to accumulate at inflamed or cancerous sites through enhanced permeability and retention (EPR) effects, a process known as passive targeting [32]. However, active targeting nanoparticles, equipped with additional targeting motifs, provide an extra layer of specificity by binding directly to targeted sites. This added precision is crucial when EPR effects are insufficient or absent, ensuring effective delivery and treatment in such cases. Active targeting nanocarriers can be tailored to treat specific diseases where overexpressed biomarkers are detected. There are many proposed benefits for this class of nanoparticles, such as reduced drug side effects, less off-site targeting, and less invasive treatment. In some cases, active targeting drug delivery systems can be paired with minimally invasive delivery systems such as IV injection, which can mitigate possible complications from alternative invasive procedures. 

Targeted nanoparticles have shown promise in various research applications. Scientists have been discovering biomarkers in various cancers, which have led to the engineering of targeting ligands that can selectively bind to cancer cells. This enables the precise delivery of therapeutic agents directly to the tumor, enhancing treatment effectiveness and reducing side effects compared to traditional chemotherapy strategies. For instance, 5-Fluorouracil was effectively delivered using nanocarriers coated with hyaluronic acid to specifically target the CD44 receptor in breast cancer [33]. Another targeting molecule for cancer is human epidermal growth factor receptor-2 (HER2) [34,35]. Additionally, a biomimetic macrophage membrane-coated nanoparticle featuring a synthetic D-form oligopeptide exhibits a strong affinity for the insulin-like growth factor 1 receptor (IGF1R), which is notably overexpressed on tumor cells [36]. Folic acid, a commonly used ligand in cancer therapy, has also been employed in various formulations such as liposomes, chitosan nanoparticles, and silica nanoparticles for cancer treatment [37,38,39].

Targeting nanoparticles are also being investigated as a promising strategy to manage cardiovascular diseases and to eliminate the complications of surgical intervention. For example, macrophage mannose receptors (MMRs) and high-density lipoproteins (HDLs) were employed for targeting macrophages at plaque sites and providing a new therapeutic strategy [40]. Nanoparticles commonly employ targeting moieties such as CD47 and Integrin α4/β1, which effectively target endothelial cells at damaged sites and facilitate successful evasion from macrophages [41,42]. EPR effects with metal particles may have higher side effects and pose a greater risk of off-site targeting. When reaching the vascular bed of an organ, small molecules can diffuse into tissues, whereas the nanoparticles are much larger and must enter cells across tissue barriers through specific uptake mechanisms. 

Targeting nanoparticles can also be used for osteoporosis to accumulate on bone regions to promote the regeneration of bone and minimize side effects, which is more beneficial compared to current medications or surgical intervention [43]. In targeting drug delivery for a condition such as osteoporosis in the elderly that does not have a blood–tissue barrier, reducing drug uptake in non-skeletal tissue is crucial [44]. Iron oxide NPs coated with dextran (Dex) and bisphosphonate (Bis) have been used for thermotherapy as treatment for osteoporosis, successfully targeting bone tissue [45,46]. Additionally, human microvascular endothelial cell (HMEC) membranes overexpressing CXCR4 have been utilized to create stem-cell-mimicking particles with targeting capabilities for osteoporosis [47]. Moreover, selective phage peptides such as SDSSD were incorporated into polyurethane nano-micelles to target osteoblasts for osteoporosis treatment [48]. Overall, targeting nanoparticles have been continuously developed as a potential treatment for various diseases to overcome the current treatment limitations. 

### 2.1. Components of Nanoparticles for Active Targeting Delivery

The targeting system, an advanced platform for drug delivery in nanotechnology, offers a diverse array of material and structural designs to efficiently deliver therapeutic payloads to desired locations as needed. The targeting nanocarrier has three main compartments: cargo, payload, and targeting ligands (Figure 1). We discuss each component in detail in the next paragraphs.

*Cargo carriers*. Extensive research in the field of nanomedicine has demonstrated that incorporating cargo to load free drugs can have many superior benefits compared to free drugs by themselves. The advantages include protecting the drug from degradation in the bloodstream, improving the drug solubility, increasing drug stability, and controlling drug release [49]. There are a variety of choices of materials that make up nanocarriers and altering the material of the nanocarrier can tune the desirability of the nanosystems. These cargos can be made from traditional synthetic materials such as polymers, silica, hydrogels, lipids, and dendrimers, or they can originate from biological sources such as virus capsids or exosomes [50,51]. Synthetic materials allow for ease in modification and optimization of the nanocarriers, while the use of nanoparticles from biological sources can create stealth effects, which can reduce immune responses to the cells. Alternatively, they can be composed of inorganic nanoparticles, such as those based on metals including silver, zinc, gold, iron, or silica [52]. Tuning these materials to a desired size can also extend the circulation time, improve cellular uptake, and enhance the permeability and retention (EPR) effects [53]. In addition, the use of nanocarriers can help co-deliver multiple agents for some combination therapies [54]. Some cargos can be made from smart materials that are responsive to a certain stimulus such as temperature, pH, enzyme, and/or ultrasound [55], to help to deliver chemotherapy drugs to tumor sites. 

*Payloads*. The second component is the drug load encapsulated or carried by the nanocarriers, which can be any therapeutic compound, including small molecules, proteins, or genetic compounds such as ribonucleic acid (RNA) and deoxyribonucleic acid (DNA) for gene therapy. Sometimes the payload cannot be delivered to the target site by traditional delivery methods such as oral delivery or injection; therefore, incorporating nanocarriers into these payloads would provide many benefits. The payloads can be toxic or may not have the capability to reach the site of interest when delivered as a free drug. For example, a chemo drug by itself does not have targeting specificity, is highly toxic, has rapid clearance, and can induce a lot of side effects in other healthy cells [56]. Currently, gene therapy is a cutting-edge approach in the field of nanomedicine. However, these genetic materials, such as DNA, RNA, ribozymes, and oligonucleotides, and the gene editing tools such as CRISPR cannot deliver themselves to the cytosol or the nuclei for transfection to occur [57]. Due to the toxicity of certain chemicals and difficulties in targeting certain therapeutics, utilizing nanoparticles as a carrier can eliminate these disadvantages. 

*Targeting ligands.* The most important component enabling this class of drug delivery systems to achieve active targeting effects is the ligands on the nanoparticle surface. Antibodies are commonly used as targeting agents incorporated into nanoparticles. For example, CD63 has led to the reduction of myocardial infarction by local delivery of exosomes or targeting CD147 to deliver doxorubicin for hepatoma therapy [58,59]. Peptides, saccharides, and small molecules represent another promising class of ligands for directing drugs to their targets. They offer advantages such as ease of modification, cost-effectiveness, and greater stability compared to antibodies due to their simple structures [60]. Antibodies are susceptible to denaturation during the conjugation process owing to chemical reactions. Notably, anti-PD-L1 peptide CLQKTPKQC, or the abbreviated D-peptide antagonist of PD-L1 (DPPA), has exhibited promise in targeted cancer immunotherapy [61,62]. Similarly, cyclic peptides such as cRGD have been shown to target integrin αvβ3 for the treatment of cancer or thrombotic diseases [63,64,65]. Aptamer, a short single-stranded oligonucleotide, can specifically bind to target molecules by forming three-dimensional structures and has been employed in nanotechnology. Recently, AS1411 aptamer has been used as a cancer targeted therapy and in magnetic resonance imaging using magnetic mesoporous silica nanoparticles [66]. DNA aptamer AP1153, which exhibits high affinity for the CCK-B receptor (CCKBR) found on pancreatic tumor cells, has been conjugated to nanoparticles [67]. This bioconjugation has resulted in increased uptake and enhanced specificity in both tumor imaging and therapeutic treatment [67].

In the new class of drug delivery systems, nanocarriers can also be coated with cell membranes that have been inherently or genetically modified with ligands [68]. While nanoparticles hold much promise, they can still be perceived as foreign entities by the immune system. To overcome this challenge and avoid immune recognition and nonspecific uptake by immune cells, researchers have shifted their focus to cell-mimicking nanoparticles. These nanoparticles capitalize on either the inherent mechanisms of cells or engineered designs. They often possess natural targeting abilities or a high concentration of adhesion molecules, enabling them to reach the affected or activated tissues more effectively and target specific sites with greater precision. For instance, T-cell membranes engineered with a chimeric antigen receptor (CAR) targeting HER2 or engineered with CD80 have been utilized to coat nanoparticles for cancer treatment, achieving the successful delivery of anticancer agents and fostering anticancer immunity [35,69]. In addition to that, erythrocyte or platelet membranes can be employed to fabricate cell-mimicking nanoparticles [70,71,72,73]. Furthermore, biomimetic nanoparticles coated with activated macrophage or neutrophil membranes have shown promise in targeting cells in the treatment of various conditions, including osteoarthritis, cancer, infectious diseases, and cardiovascular diseases [74,75,76].

Taken together, these three key components for active targeting delivery, that is, the carrier, payload, and targeting ligands, can be combined into a rich variety of possible therapeutic vehicles, as summarized pictorially in Figure 1. These targeted nanoparticles enhance therapeutic effects locally and specifically to the targeted sites for a wide range of therapeutic applications, including cancer, cardiovascular, and infectious diseases.

### 2.2. Current Barriers in Translating Targeted Nanoparticle Delivery to the Clinic

Although targeting nanoparticles demonstrate superior potential in drug delivery, no targeting nanoparticles have been reported for clinical use due to many possible reasons. One of them is that nanoparticles, especially biological nanoparticles, do not have standardized processing and characterization methods. Quality control of nanomaterials can help to achieve the repeatability, determine progress efficiency, examine the activity, and ensure the safety of the product [15]. Better quality control of nanoparticles can lead to more homogenous NP formulations that could result in better outcomes in vitro and in vivo [16,17]. However, current quality control for nanoparticles has no standardized physicochemical characterization [18] and relies on ensemble-average (bulk analysis) studies, which do not provide in-depth information about possible inconsistencies in drug loading and surface coating during NP production. A case study showed that the batch-to-batch inconsistency in the PEGylation coating of gold nanoparticles results in different results in vivo [19]. Another study showed that different traditional techniques that have been used to record the amounts of drug content are not consistent and can result in inconsistencies in reported data [20].

In general, the process of generating nanoparticles with adequate drug loading and targeting moiety coating is complicated, and it is still challenging to assess the quality of these nanoparticle formulations. Currently, we do not know how many ligands can be attached per nanoparticle, nor do we know the fractions of coated and uncoated NPs, or even partially coated nanoparticles, per solution. Adequate quality control can help minimize the dosage of materials and drugs used and maximize the efficiency of drug delivery systems, thereby also reducing possible drug side effects. There is a need to strengthen the characterization methods that can push the translation progress of targeted nanoparticles [21].

## 3. Methods to Quantify Sizes of Nanoparticles

Standard methods for routinely characterizing nanocarriers for drug delivery applications investigate parameters such as size, surface charge, and morphology [49]. Nanoparticles’ size and size variability can directly influence the behavior, efficacy, and safety of the drug delivery vehicles. A broad range of nanoparticles (different sizes of nanoparticles) may lead to inconsistent drug distribution and different uptake pathways. It is crucial to study nanoparticles in colloidal systems, such as nanoparticles in suspension, to understand their stability, aggregation tendencies, and interactions in biological environments. Standard characterization methods provide information on those attributes by measuring the size distribution and zeta potential [77]. The size of nanoparticles is important because it determines where the nanoparticles can be deposited in the body. It affects cellular uptake, biodistribution, and the EPR effect, which is particularly relevant for the passive targeting of tumors and inflamed tissues with leaky vasculature [78].

Nanoparticle size is also critical for avoiding biological barriers, such as kidney filtration; nanoparticles smaller than 10 nm are usually rapidly filtered by the kidneys [79]. Additionally, size influences the immune system’s responses to nanoparticles [80]. For instance, nanoparticles with a small size (<10 nm) can be recognized by immune cells, and larger particles (>200 nm) can be easily engulfed by immune cells [81]. The zeta potential also affects the biodistribution of nanoparticles in the body. Nanoparticles with a stable zeta potential are less prone to aggregation and more likely to circulate in the bloodstream. Nanoparticles with an unstable zeta potential lead to a rapid clearance in the reticuloendothelial system (RES) [81]. The charge of nanoparticles can also affect nanoparticle uptake into cells. Nanoparticles with a charge above 30 mV and lower than −30 mV are stable; nanoparticles within this range have a strong repulsive force that prevents aggregation [82]. A stable zeta potential of nanosystems is crucial to ensure that nanoparticles can be uniformly dispersed, in order to enhance their bioavailability, targeting efficiency, and controlled release properties.

### 3.1. Bulk Detection

One of the most common and routinely used techniques to quantify the size distribution of nanoparticles is through dynamic light scattering (DLS), which is also known as photon correlation spectroscopy (PCS) [83]. The DLS method is used to quantify nanoparticle size and can be used to quantify the stability of nanoparticles over time [84]. Based on particle Brownian motions, DLS can measure the fluctuation of light intensity and determine the diffusion coefficient (which correlates to the hydrodynamic radius) [84]. However, dynamic light scattering methods are not efficient in distinguishing nanoparticles in polydisperse solutions [15]. Since the intensity of the scattered light used in DLS is proportional to the sixth power of the particle diameter, this method is more sensitive to large-size particles [85]. The size ratio in polydisperse solutions limits size differences so that they cannot be accurately detected from a range of 2:1 to 3:1 because the diffusion coefficient is inversely proportional to the radius and the particle fluctuations cannot distinguish particles of similar sizes [86].

Small angle X-ray scattering (SAXS) has been used to study nanoparticles since the 1950s, and this technique can be used to determine nanoparticle size, size distribution, shape, and surface structure [87]. SAXS can measure the size of particles by the radius of gyration (using Guinier’s law), particle surface area, and the scattering variance [88]. SAXS has a working range from 1 nm to 1000 nm [89]. In SAXS, a collimated, monochromatic X-ray beam hits the sample, and the detector records the scattered radiation at low angles (typically a few degrees) [89]. SAXS has been used to determine nanoparticle properties in a variety of settings such as aerosols, colloidal suspensions, powders, solids, and thin films [90], while the sample preparation methods are considered simple. Compared to electron microscopy (EM) methods, the SAXS technique produces more statistically reliable results for particles sizes as it can measure a larger population of nanoparticles more quickly [87,90].

Another technology that can be used for nanoparticle characterization is centrifugal sedimentation, also known as differential centrifugal sedimentation (DSC), or centrifugal photo-sedimentation [91,92]. DSC is a method that is based on centrifugal force (also known as settling velocity) to determine the size of the nanoparticles. Nanoparticles with larger sizes will settle faster, while the smaller-sized particles tend to settle slower. The traveling distance of the particle is measured from the initial position to the detection position of the nanoparticles, and the settling velocity is calculated. Based on Stokes’ law (with the assigned parameters of particle density, solvent density, and solvent viscosity), the particle size can be determined. There are two modes: line start mode and homogenous mode [91]. In line start mode, a sample is introduced into the density gradient solution, and the size of the particle is then calculated from the settling time that can reach the detector, which allows it to measure a small quantity of high-concentration solution with high resolution. In homogeneous mode, the settling time will start from a uniformly distributed sample, and the particle size distribution can be calculated based on information of the particles through the detection zone [91]. Homogeneous mode is suitable for measuring samples with low concentrations.

Another technique that can be used for characterizing NPs is size-exclusion chromatography (SEC), which can be used for analyzing the size distribution of nanoparticles; it has mostly been used to characterize, exosomes, LNPs, and viral vectors [93,94]. Based on the size of the nanoparticles, SEC will separate them based on the difference in their hydrodynamic radius. The analytes will flow through a column, where larger molecules will travel faster in the column and smaller molecules will travel slower in the mobile phase. Furthermore, it can also be coupled with detectors such as mass-spectrometry (MS) and multiangle light scattering (MALS) to enhance the sensitivity [93,95].

Nanoparticle distributions can also be studied by using asymmetric flow field-flow fractionation coupled with multiangle light scattering and dynamic light scattering (AF4-MALS-DLS) [96,97]. This approach allows for the assessment of batch-to-batch variability and changes in the size of nanoparticles induced by serum [96]. There is a strong need to raise awareness of the limitations of batch mode DLS and a need to develop a more reliable approach for the evaluation of particles in simple and complex media [96]. It is reported that AF4-MALS-DLS provides a higher resolution than batch mode [96,97].

### 3.2. Single-Nanoparticle Detection

Tunable resistive pulse sensing (TRPS) is a quantification technique that allows particle-by-particle detection [98,99]. Colloidal particles are resuspended in a conductive solution, and each nanoparticle passes through single or multiple pores in a membrane. Electric potential is applied across the membrane, and when a nanoparticle passes through the pore, it results in a drop in current (which is called a resistive pulse). The height, width, and frequency of these resistive pulses provide information on the size, surface charge properties, and total concentration of the analyte [98]. TRPS is one of the few nanoparticle characterization techniques that can provide sensitive and high-resolution measurement to study particle properties in biological environments [98].

Electron microscopy (EM) methods such as Transmission Electron Microscopy (TEM) and Scanning Electron Microscopy (SEM) can be used to study the morphology (direct measurement of geometrical size) and size distribution of nanoparticles [83,100]. In SEM, a focused beam of electrons is used to scan the sample surface, which results in a three-dimensional image and surface texture. The resolution of SEM is 3–20 nm [83]. TEM uses a beam of electrons to transmit through a specimen, which results in images of the interior of the samples. The resolution of TEM is 0.2 nm; however, the samples have to be processed before the resolution can be changed to 2 nm [83]. Depending on the properties of samples, some samples may need to be fixed on the grid and negative staining applied with heavy metal salts such as uranyl acetate to enhance the contrast of the samples. Another more advanced type of TEM method, cryogenic TEM (cryo-TEM), has a similar working principle to regular TEM and can help to preserve the structures of nanoparticle samples while imaging. Cryo-TEM is a preferred EM technique to visualize the close to native state of nanoparticles, as this technique does not require the sample dehydration step, which can be more suitable for liposomes, LNPs, and exosomes [101,102,103,104]. Overall, both SEM and TEM provide single-molecule characterization and high-resolution characterization, but they have different throughputs. EM methods only provide a few hundred to a few thousand particles for measurement [90]. This method is time consuming and can be affected by bias when obtaining data about size distribution [83].

Nanoparticle tracking analysis (NTA) is another single-molecule analysis technique that can measure the particle size distribution for polydisperse samples [105]. NTA was commercialized in 2006 and can detect particles in the range of 30 to 1000 nm [85]. The lower detection limit of 30 nm depends on the refractive index of the particle [85]. In NTA, a microscope with an attached scientific grade video camera can track the illumination of particles undergoing Brownian motion in solution [105]. This allows the determination of the distribution of nanoparticle sizes, which is an advantage over the ensemble methods such as DLS [105]. However, there are limits to NTA in distinguishing particle sizes based on the Einstein–Stokes relation alone, as it is challenging to resolve the peaks in more complicated samples such as quadrimodal ones [106]. Additionally, the nanoparticle concentration is more critical in NTA than DLS because NTA has a more narrow concentration range that can be detected: too low of sampling will lead to poor statistical sampling, and too high of sampling will lead to nanoparticles overlapping in the measuring window [85].

Atomic force microscopy (AFM), which was introduced in 1986 for characterizing nanoscale objects, can also be used to measure the sizes of nanoparticles [86]. In AFM, a cantilever-shaped probe is used to scan the surface of a specimen [107]. AFM uses piezoelectric ceramics to change the direction of scanning of a specimen in the X, Y, and Z directions, and the AFM tip is placed above the specimen so that it is either repelled or attracted by the forces of the surface of that specimen [86]. The bending movement of the cantilever changes the direction of incident laser light that is deflected by it and collected by a photo-detector [107]. The detector signal amplitude is then processed through a feedback loop, and data acquisition software turns the measured cantilever deflections to a 3D image [86]. Different from SEM, which only allows two-dimensional images of nanoparticles, AFM allows for acquiring the three-dimensional surface profile of a particle [107]. AFM can help to identify a large bimodal size distribution compared to DLS, which measures only a single peak [86]. Since the samples are deposited on the substrate via various deposition techniques, AFM scans cannot always calculate the ratio between the diameters of the particles [86]. A summary of standard single-nanoparticle analysis methods and their relative advantages and disadvantages is presented in Table 1.

## 4. Methods to Quantify the Drug Loading Efficacies of Nanoparticles

Nanocarriers provide a versatile platform capable of encapsulating a wide range of compounds, including DNAs, proteins, and various biomolecules. In the case of targeted delivery, after the nanoparticles are delivered to the site of interest, the nanoparticles will release the drug load in a controlled manner to the intended cells and show therapeutic effects. However, the amount of drug successfully loaded to the nanocarriers can be different from the intended amount. Most of the current drug delivery vehicles have the drawbacks of having low drug contents (less than 10%) with higher contents of polymer carriers [110]. Additionally, the extensive use of polymer carriers can lead to systemic toxicity [110]. Therefore, researchers want to optimize the amount of drug loaded inside nanocarriers while also keeping it consistent to control the dosage and therapeutic window. 

### 4.1. Bulk Detection of Payloads in NPs

Currently, the most common method to quantify drugs loaded onto nanoparticles is through bulk detection, and there are two ways that one can use to determine drug loading: indirect and direct. Many studies have been conducted to compare the reliability between these two different methods [20,111]. The indirect method can quantify the content that remains in the supernatant during synthesis. This means the content of the drug that is detected on the byproduct will be quantified, and by subtracting from the initial amount of drug used for the synthesis, we can estimate the amount that is in the vesicles. The second traditional method is direct quantification. This process requires isolation, where the nanoparticles will be isolated into the drug contents and drug carriers. Then, the isolated drug content is quantified using existing drug detection methods.

*Indirect drug loading quantification*: In this approach, the amount of unincorporated drug from the NP synthesis process is quantified. For example, in the process of making synthetic PLGA nanoparticles, the drug is added, and after that, the nanoparticles and the surfactant solution are partitioned by ultracentrifugation. The nanoparticles with the drug incorporated are isolated from the supernatant with the unincorporated drug, the supernatant is then collected, and the drug amount is quantified. There are, however, some factors that can affect the reliability of this drug quantification such as a very heterogeneous preparation medium and interactions between the drug and surfactant [20]. In this indirect approach, the drug loading content and loading efficiency are calculated by Equations (1) and (2). (1)$$Drug\;loading\;content\left(weight\;\left[wt\right]\%\right)=\frac{\left(Mass\;of\;initial\;drug-mass\;unincorporated\;drug\;amount\right)}{Amount\;of\;nanoparticles\;recovered}\times 100$$(2)$$Drug\;loading\;efficiency\left(\%\right)=\frac{Intial\;mass\;of\;drug-mass\;of\;unincorporated\;drug\;amount}{Total\;drug\;used\;in\;formulation}\times 100$$

Overall, the indirect method tends to be preferred by users due to its less complicated procedures relative to the direct method. Based on various studies [110], the indirect loading method tends to estimate higher loading amount values than direct methods and is considered to be less accurate than direct loading methods. 

*Direct drug loading quantification:* In this approach, the nanoparticles can be dissolved in an appropriate organic solvent, where the drug will not be dissolved [112,113]. The precipitating drug is then redissolved in an appropriate buffer, and the amount of drug loading is then quantified via common drug detection methods [112]. For this strategy, the drug loading content and drug loading efficiency can be calculated based on Equations (3) and (4) [110].

There are some challenges in using this direct method such as the requirement to completely dissolve the nanoparticle polymers, completely redissolve the drug, and possible interactions between the drugs and surfactants [20]. Direct methods are more complicated than the indirect method, but they are more accurate. However, based on different detection techniques, recorded values vary, and accuracy is still a challenge [20].(3)$$Drug\;loading\;content(weight\;\left[wt\right]\%)=\frac{Mass\;of\;the\;drug\;in\;nanoparticles}{Initial\;mass\;of\;nanoparticles}\times 100$$(4)$$Drug\;loading\;efficiency\left(weight\;[wt]\%\right)=\frac{Actua\;lmass\;of\;drug\;in\;nanoparticles}{Mass\;of\;drug\;used\;in\;formulation}\times 100$$

*Quantification methods for detecting therapeutic payloads:* Irrespective of whether a researcher uses a direct or indirect approach, as described above, they will need to quantify the amount of drug or genetic load in a free solution. In the indirect approach, it will be the therapeutic load released during synthesis or unintentional breakdown of nanoparticles, whereas in the active approach, it will be the load of intentionally broken nanoparticles. 

In the case of drug loading, the nanocarrier can be separated from the drug by isolation methods such as centrifugation, dialysis, ultrafiltration, solvent-extraction, precipitation, high-performance liquid chromatography, field flow fractionation, and analytical centrifugation [35,114]. Then, the samples containing the drug can be quantified by analytic tools such as biological assays, UV spectrometry (UV/VIS), or chromatography [35,114,115].

Genetic loads of nanoparticles typically involve nucleic acids such DNA and RNA to be incorporated into the nanocarriers. Specifically, they can use antisense DNA, messenger RNA (mRNA), small interfering RNA (siRNA), microRNA (miRNA), or plasmid DNA [116,117]. Gene delivery systems in nanotechnology are actively being developed, employing cationic polymers, polysaccharides, poly(ethylenimine)s, and poly(L-lysine) to enhance the cellular uptake of DNA systems and facilitate transfection [116,118,119]. Alternatively, genetic materials can be added to biomimetic nanoparticles such as LNPs and exosomes, which, due to their ability to fuse with the cells better, have better endosomal escape, lower toxicity, and higher compatibility. Quantifying the nucleic-acid-based drug content is crucial for assessing therapeutic efficacy. Popular methods include UV absorbance and fluorescent-based assays. UV/VIS or Nanodrop spectroscopy can quantify nucleic acids through the former method, while the fluorescent assays, such as the Picogreen assay, utilize fluorescent detection of Picogreen dyes, which bind to double-stranded DNAs (dsDNAs), forming a highly luminescent complex. However, degradation may render nucleic-acid-based drugs ineffective, and absorbance or fluorescent-based assays do not distinguish between effective and ineffective forms. Gel electrophoresis is a standard method for quantifying DNA and DNA fragments based on weight while providing quantitative assessments of nucleic acids and evaluating their functionality [113,120].

The development of peptide and protein therapeutics has been significant in recent decades, yet their delivery poses several challenges including denaturation, degradation, low bioavailability, and metabolic instability. To address these limitations, nanotechnology plays a crucial role in enhancing protein delivery. Examples of protein-based therapeutics currently available include Etanercept, Insulin glargine, Peg filgrastim, Cyclosporin, Octreotide, and Liraglutide [118,121]. Oral delivery of proteins faces obstacles such as degradation in the gastrointestinal tract and the inability to cross the epithelial layer. Intravenous (IV) and subcutaneous (SC) delivery methods can overcome the absorption barriers; however, they present challenges such as systemic protease activity, rapid metabolism, opsonization, conformational changes, and dissociation of subunit proteins [118]. Protein-based nanotechnology has emerged as a promising alternative to address these challenges [122,123]. UV absorbance at 260 nm is commonly used to quantify nucleic acids due to DNA’s absorption peak at this wavelength. However, polymeric degradation during drug release may affect absorbance values. Protein quantification at 280 nm relies on specific absorption by ring amino acids such as tryptophan and tyrosine. While this method is simple, quick, and reproducible, it may be inaccurate for peptides lacking these amino acids and can be influenced by background factors such as coexisting DNA or degraded polymer. The Bicinchoninic Acid (BCA) assay quantifies proteins based on their ability to reduce Cu^2+^ to Cu^1+^, resulting in a purple color formation. However, this assay is susceptible to interference from certain chemicals such as reducing agents, copper chelators, and high-concentration buffers. The mechanism of this dye-based protein detection is based on the binding of protein molecules on Coomassie dyes under acidic conditions. Proteins can be quantified based on the presence of basic amino acid residues such as arginine, lysine, and histidine. These amino acids will contribute to the protein–dye complex. At low concentrations, the reducing agents will not cause interference, which results in a lower detection range compared to BCA assays. However, the presence of SDS (even at low concentrations) can affect the reading [124,125]. Mass spectrometry (MS) is a technique used to quantify proteins and peptides by analyzing their amino acid composition. Each amino acid is ionized, and the resulting ions are separated based on their mass-to-charge ratio. A summary of bulk detection methods to quantify the average drug or genetic load per nanoparticle is shown in Table 2.

Additionally, there are also many bulk-detection assays and methods that can be used for application for drugs such as amine derivation (detection range: 0.05–25 µg) and detergent-based fluorescent detection assays for protein-based analytes [126]. Methods such as capillary electrophoresis can help to determine the nucleic acid integrity and degradation [127,128]. However, using gel-electrophoresis is a semi-quantitative result, with time-consuming procedures and a low sensitivity [129]. Nevertheless, new methods such as coupling capillary electrophoresis and laser-induced fluorescence detection can also be used to determine drug load detection based on Sciex company, which has been used to detect mRNA loading on LNPs. More quantitative polymerase chain reaction (PCR) methods such as digital PCR (dPCR), quantitative PCR (qPCR), and reverse transcription PCR are more sensitive and quantitative, showing promise for use in quantifying payloads.

**Table 2 bioengineering-12-00362-t002:** Available bulk detection methods for determining drug or genetic loading.

**Payload**	**Detection Methods**	**Detection Range**	**Advantages**	**Disadvantages**	**Application Examples**
Protein based	UV Absorbance 280 nm	20–3000 µg [126]	Simple, highly specific, sample can be used after measurement	Applicable to proteins with tryptophan and tyrosine; requires non-sequence-specific absorbance calibration and accounting for nucleic acid background noise.	Nerolidol-Loaded Chitosan–Alginate Nanoparticles [130]
	UV Absorbance 205 nm	1–100 µg [126]	More sensitive and displays less protein-to-protein variability than 280 nm [131]	Background of solvent can interfere with the reading.	N/A
	Coomassie Blue (Bradford Assay)	1–50 µg [126]	Simple	Contaminate from surfactant.	Platelet-rich-plasma-loaded chitosan nanoparticles (wound healing) [132], tumor-antigen-loaded PLGA NPs (cancer vaccine) [133]
	Lowry (Alkaline Copper Reduction Assays)	5–100 µg [126]	Highly sensitive	Long preparation and complex procedure, contaminated by reduction reaction.	BSA-Loaded PLGA–Chitosan Composite Nanoparticles [134]
	Bicinchoninic Acid (BCA)	0.2–50 µg [126]	Simpler, highly sensitive	Thiol, phospholipid, and ammonium sulfate interference.	PLGA-containing anti-CTLA4 (endometriosis) [135] PLGA-R837@Cat nanoparticles (tumors) [136]
	Mass-spectrometry (MS)		Flexible, specific, multiple targets simultaneously, precision and accuracy, requiring minimal material for analysis [137]	High equipment costs; stable electricity, ventilation, high-purity gases, and skilled staff are required. Compounds must be volatile enough to transfer from liquid to mobile carrier gas for detector elution [137].	Docetaxel-loaded PLGA NPs (cancer) [138]
Nucleic acids	UV Absorbance 260 nm	NanoDrop 1/1: 0.2 *–27,500 ng/µL Nanodrop 3300: 0.05–2000 picograms/μL (Thermofisher, Waltham, Massachusetts, USA)	Quick and easy	Cannot differentiate between RNA and DNA, limited sensitivity at low concentrations.	DNA-loaded PLA-PEG-PLA NPs (SS-Nanodrop) [139] DNA/PLL NPs (Binding Kinetics) [140]
	Fluorescence-based assays: Picogreen assay (DNA) Ribogreen assay (RNA)	1 ng/mL to 1000 ng/mL (Picogreen assay) 1 ng/mL to 1 µg/mL Ribogreen assay—RNA) [141] iQuant RNA BR Assay Kit (RNA): 20–1000 ng RNA	Higher sensitivity and specificity [129]	Susceptibility to compounds such as salts and chemical reagents [129].	cDNA PLGA NPs-Notch SIgnaling [142] IL-29 cDNA-immunology (HCV, cancer) [143] RNAi PEI-PLGA for gene delivery [144] Various types of RNA-loaded LNPs [141] Reactive oxygen species (ROS) mRNA LNPs [145] mRNA LNPs for prenatal treatment of congenital disorders [146] mRNA-encoded cystic fibrosis transmembrane conductance regulator (CFTR)-loaded LNPs for pulmonary delivery [147] LNPs incorporating hydroxycholesterols to enhance mRNA delivery to T cells [148]
	Liquid chromatography–mass spectrometry (LC-MS)	used for measuring mRNA	Sample does not require mRNA extraction, detergents, or enzyme	Expensive, limited sample throughput.	mRNA loaded in lipid nanoparticles (LNPs) [149]
Other pharmaceutical compounds	High-performance liquid chromatography (HPLC)	190–800 nm	High sensitivity, specificity, rapidity, accuracy, precision, and ease of automation	Expensive, complicated to troubleshoot, and time-consuming.	DOX-loaded liposomes (cancer) [150], Isoniazid- and Rifampicin-Loaded Bovine Serum Albumin Nanoparticles (tuberculosis) [151], melatonin-loaded human serum albumin NPs (neurodegenerative eye diseases) [152]
	UV/VIS	variable	Easy to use, fast and efficient analysis, inexpensive, non-destructive, minimal processing	Lower sensitivity and selectivity; light scattering and multiple absorbing species may interfere with accuracy.	Itraconazole-loaded chitosan-silver nanoparticles [153] necrosulfonamide-loaded mesoporous nanoparticles (inflammation) [154] Cisplatin-loaded Glutathione-responsive biodegradable polyurethane nanoparticles (cancer): UV–Vis spectrophotometer at 703 nm [155] Curcumin-loaded self-assembled WPI@SLG core–shell nanoparticles (antioxidant activity in gastrointestinal conditions): absorbance at the wavelength of 426 nm [156] Ciprofloxacin-loaded PEG–PLGA NPs (regenerative endodontic treatment): absorbance 275 nm [157] ATP-loaded albumin nanoparticles (cancer): UV/Vis spectrophotometer at 257 nm [158]
	Optical Density (OD)	Variable	Easy to use, inexpensive	Low sensitivity.	Ponatinib- and dasatinib-loaded exosomes (cancer): The optical density (OD) value for ponatinib and dasatinib was recorded at 285 and 233 nm [159]

### 4.2. Single-Nanoparticle Therapeutic Load Detection

Single-nanoparticle analysis is currently a less common way to detect the amount of therapeutic payload. Most of the analytes characterized this way to date are extracellular vesicles (EVs) and LNPs [160,161]. There is still a gap in quantitative methods to learn about the payload. Questions such as how much payload is loaded in a single particle remain unanswered. 

Single-molecule localization microscopy (SMLM) is a super resolution imaging approach that was used to quantify the copy number of GFP molecules loaded in single extracellular vesicles [162]. Another method using fluorescence called nanoflow cytometry (nFCM) has been used to quantify the amount of protein loaded in EVs [163]. This technique can detect nanoparticles and measures both fluorescence signals and scattered light (forward-scattered light and side-scattered light). nFCM overcomes the limits of conventional flow cytometry through technical optimization. Due to its ability to detect single molecules, nFCM can analyze the heterogeneity of GFP-loaded EVs. nFCM can detect single EVs that are less than 40 nm and DNA fragments that are 200 bp [163]. nFCM can detect nanoparticles in the range of 7 nm [163]. Molecular beacons have also been used for the detection of EVs [164]. Molecular beacons are oligonucleotide probes that can have both fluorophore and quencher moieties. They can be used to detect specific nucleic acids. They have the complementary sequence to a specific DNA or RNA. When they bind to a DNA or RNA complementary sequence, they undergo conformational changes that cause the fluorophore get to separate from the quencher, thus enabling fluorescence emission [164]. One obvious limitation of all the above-described single-particle content analysis methods is that they cannot perform label-free quantification of therapeutic loads. 

Digital quantitative polymerase chain reaction (dqPCR) combines single-nanoparticle analysis using microfluidic digital PCR to measure the internal cargo of individual liposomes. This method is a combination of two orthogonal polymerase chain reaction (PCR) techniques: digital PCR (dPCR) and quantitative PCR (qPCR). The qPCR dimension quantifies liposomes to validate their capture in the single-liposome analysis regime. The qPCR dimension quantifies DNA copy numbers packaged within the liposomes. qPCR can be used to quantify nucleic acid loading in lipid vesicles. The nucleic acid copy number is determined from the quantitative cycle (Cq) measured from the qPCR curve, but some of the limitations are that the liposomes need to be lysed and that the analysis is performed in bulk. Digital PCR (dPCR) is an amplification technique that measures nucleic acids with single-molecule sensitivity. qPCR-dPCR divides a sample into >10^3 individual low-volume PCR reaction partitions (droplets or chambers). dPCR can be used to detect liposomes but cannot be used to detect liposomal cargo [165]. Combining both techniques allow the capturing of single nanoparticles for analysis.

Cyclindrical illumination confocal spectroscopy (CICS) coupled with single-nanoparticle free solution hydrodynamic separation (SN-FSHS) is a new and high-throughput technique to study hydrodynamic size and RNA loading. It is capable of revealing the heterogeneity of the packing density. The first component of the system, SN-FSHS, is a hydrodynamic chromatography (HDC) technique to partition the nanoparticles based on size due to the different migration velocities, and it can reveal information about the hydrodynamic radius and the size distribution of the NPs. The second component of the system, CICS, utilizes one-dimensional beam shaping, which helps allow the single-molecule characterization of fluorescent-tagged components [166,167]. This technique has been used to study siRNA-loaded LNPs.

## 5. Methods to Quantify the Number of Targeted Ligands per Nanoparticle

There are many methods to modify nanoparticle surfaces to enable targeting for specific applications. Nanoparticles can be functionalized with ligands such as antibodies, peptides, and aptamers through different conjugation chemistries [6,168,169,170]. Some nanoparticles can be coated with biomimetic membranes that have ligands of interest. Ligand types, conjugation strategies, ligand orientations, and densities are the crucial parameters associated with active targeting nanoparticles [171]. However, ligand density is the parameter that is usually underestimated [171]. A higher ligand density per particle can lead to a higher probability of binding to the target [171]. However, a higher targeting ligand density can also have some disadvantages at the particle, cellular, and systemic levels [171]. At the particle level, a high ligand density can lead to an increased size, lower diffusion coefficient in biological matrices, and lower colloidal stability. At the cellular level, a high ligand density can lead to steric hindrance that inhibits receptor binding or a high consumption of cellular receptors per binding event. Lastly, at the systemic level, a high ligand density can lead to decreased stealth characteristics, increased opsonization, and increased RES clearance [171]. Although there are reported methods to help control the number of ligands per nanoparticle, there is no easy method for ligand quantification [171]. We will discuss more specifically the current methods that rely mostly on bulk detection and the more innovative single-molecule methods have been developed to quantify the density on each single particle. 

### 5.1. Bulk Detection of the Average Number of Ligands per Nanoparticle

The most common method to quantify the conjugation density is through bulk detection, which will roughly quantify the amount of conjugates/coating, and the average number of ligands per nanoparticle can be estimated. In bulk detection, there are two methods to confirm the conjugation density, through direct or indirect methods. The indirect method quantifies the content of ligand that remained in the byproduct solution during the bioconjugation or coating process. The byproduct solution and the conjugated product will be separated via various separation techniques such as centrifugation. The content of the unbound ligand in the byproduct solution will be quantified and will be subtracted from the initial amount of ligand added. The other method is direct quantification, where the conjugated nanoparticles can be directly measured.

Most of the reported research quantifications are based on indirect measurement. Nanoparticles are usually incubated with ligands in a solution that allows the conjugation to happen. The population of nanoparticles and the bound conjugates are separated from the unbound conjugates. The unbound conjugates in the remaining solution will be quantified using quantitative methods such as direct UV/VIS or with colorimetric assays (BCA assay, Bradford assay), HPLC, etc. Indirect quantification of the conjugation, such as with the BCA assay, usually results in an overestimation of conjugates on nanoparticles [172,173]. The traditional method for quantifying targeting ligands may not be suitable for biological nanoparticles such as liposomes, as they cannot withstand harsh separation techniques. Therefore, direct quantification would be a more desirable method for quantifying loading. 

Fluorescence-based methods have been used to directly quantify antibodies immobilized on gold nanoparticles. Antibody-modified gold nanoparticles were treated with potassium iodide (KI) and iodine (I_2_) etchant to dissolve the gold nanoparticles. A desalting spin column was used to recover the antibody released from the nanoparticles, and NanoOrange, a fluorescent dye, was used to quantify the antibody. We determined that 309 ± 93 antibodies adsorb onto a 60 nm gold nanoparticle (2.6 × 1010 NP mL) [173].

Enzyme-mediated assays can be used to quantify the fraction of immobilized antibodies that arcche accessible for antigen binding. Anti-horseradish peroxidase (anti-HRP) antibody is mixed with AuNPs to allow for conjugation, and the unbound, excess antibody is quantified with a modified Bradford assay to determine antibody loading onto AuNPs. The conjugates are then mixed with excess HRP to saturate all accessible binding sites, and bound HRP is quantified based on the enzymatic reaction rate [174].

NMR spectroscopy can be used to confirm ligand immobilization on nanoparticles [175,176]. It can provide both quantitative and qualitative data that can be used to characterize the successful modification of nanomaterials by ligands, differentiate between bound and unbound ligands, and quantify bound ligands. NMR data can help in understanding the ligand binding mode and dynamics of the bound ligands and in studying the interactions of surface-functionalized nanomaterials with biomolecules to help infer ligand structure [175].

IR spectroscopy can also be used to infer ligand structure [175]. In the analysis of nanomaterial surface functionalization, IR spectroscopy is primarily used to confirm the functionalization of nanoparticles by comparing to the spectrum of the free ligand [175]. There are several other characterization methods that can be used for quantifying ligand structure and confirming its presence on nanoparticles such as Surface Enhanced Raman Spectroscopy (SERS), Scanning Tunneling Microscopy (STM), Small-angle neutron scattering (SANS) microscopy, and X-Ray Diffraction Spectroscopy (XRD) [175]. XRD can not only analyze ligand structure and conformation but can also map the position of the surface ligand on nanoparticles. Other tools that can be used to determine the ligand density of nanoparticles are Thermogravimetric Analysis (TGA), Quantitative NMR (qNMR) ligand density [175], and Time-of-Flight Secondary Ion Mass Spectrometry (ToF-SIMS) [175].

### 5.2. Single-Nanoparticle Ligand Density Detection

There is a need to measure nanoparticles in which the numbers of targeting ligands and receptors can specifically be counted with single-molecule resolution to understand their interaction [177]. TEM and SMLM have been used to quantify the number and map the position of functional sites of proteins on the surface of single nanoparticles [178,179,180].

TEM can be used for quantifying the size and the loading; it can also be used for confirming the bioconjugation [181]. The conjugation between gold nanoparticles (AuNPs) and Cetuximab (antibodies to treat cancer) was visualized by TEM [178]. 5-Aminolevulinic acid conjugated to AuNPs was observed under TEM due to their slight change in size [182]. TEM is also helpful for characterizing the orientation of covalently conjugated proteins on nanoparticles [178]. Specifically, silica nanoparticles were conjugated with transferrin, and antibody fragments (Fabs) conjugated with gold nanoparticles were used as probes to map any available epitopes on transferrin [179]. TEM has also been used to confirm the coating efficiency of membrane-coated nanoparticles [183,184].

Direct stochastic optical reconstruction microscopy (dSTORM) is an SMLM technique that has a spatial resolution of 20 nm that can also help determine the number of antibody conjugates per nanoparticle [177]. In one study, silica nanoparticles with different sizes (50, 100, 150 nm) were conjugated with Cetuximab, which is an EGFR antibody conjugated to a dSTORM probe (EGFR-AF647) through incubation. Analysis of the acquired dSTORM images helped quantify differences in the density of antibodies on each nanoparticle [177]. Additionally, (DNA-PAINT) is another SMLM technique that has been used to map the conjugates attached onto nano- and microparticles [180] by using a qualitative PAINT (qPAINT) algorithm. Nanoparticles conjugated with Streptavidin and microparticles conjugated with Digoxigenin antibodies have been used to map the ligand distribution on them [180]. For this procedure, conjugated nanoparticles were incubated with a PAINT probe that has the ligand attached to the conjugate and a DNA docking strand, which allows the probe to bind specifically to the conjugates. Then, the imager DNA strand, which is a complementary DNA strand to the docking strand, was added. The transient binding between the docking strand and the imager strand resulted in the imaging and localization of single ligands. The results of these analyses showed that particles with a similar size can have considerable variability in the number of conjugates [180].

## 6. Summary and Future Outlook

The field of nanomedicine is rapidly advancing, and as the complexity of nanoparticles grows, more sophisticated technologies will need to be developed for characterizing these nanoparticles. There are still many gaps in standardizing the characterization of heterogeneous formulations for particle therapeutic loading and targeted moiety coating. Several key shortcomings of bulk solution analysis technologies are summarized in this work. These shortcomings are beginning to be addressed by emerging single-nanoparticle characterization technologies. However, there is still a lot of space for improvement, as well as room for opportunities for new methods to expand into this space. There are currently no methods to quantify both nanoparticle loading and coating at the single-nanoparticle resolution at high throughputs. It is also not clear how this information, once attained, could be actionable for improving nanoparticle preparation heterogeneity. It is also not clear what level of detail needs to be known about preparation before that knowledge results in successful translation into pre-clinical trials and human use. Nevertheless, the value of these new methods is potentially immense and is likely to be crucial for scientists to improve the quality of nanoparticle preparations in a quantifiably reproducible manner. 

The following are emerging technologies that can help characterize single-nanoparticle characterization.

*Nanopore-based technologies:* Label-free, single-molecule nanopore sensing has been expanding for the sequencing of nucleic acids and protein-based analytes [185,186], but it has recently also shown potential for characterizing nanoparticles [187,188]. In this technique, nanoparticles pass one at a time through a nanopore drilled into a thin dielectric material. The pore diameter is made to be slightly larger than that of the nanoparticles, but not much larger, so that during transit of the nanoparticle through the pore, the pore current is transiently blocked, resulting in a resistive pulse detected by a high-sensitivity voltage clamp system. The recorded fluctuations in current time-series data can be analyzed to provide indirect information on nanoparticle size, shape, surface charge, and other properties. As this technology can detect and analyze individual nanoparticles, it can assess heterogeneity within nanoparticle populations. This information can be beneficial to drug delivery applications, where understanding the uniformity and specific attributes of nanoparticles can significantly impact their therapeutic performance. Additionally, this is a label-free method, avoiding the need for additional chemical modifications that could alter nanoparticle behavior. Therefore, expanding this technology for studying nanomedicine can provide a powerful tool for rapid screening and quality control at a single-nanoparticle resolution.

Plasmonic nanopores represent a more recent evolution in nanopore technology, where optical measurements can be collected concurrently with electrical signals during nanoparticle translocation through the pore. Plasmonic enhancement of light intensity can help increase the fluorescence emission of labeled particles, or it can be used for both label-free sensing and optical trapping [189]. In more recent work, plasmonic optical sensing has been combined with alternating current (AC) electrical sensing [188]. The premise of the latter is that a nanoparticle is considered as a spherical capacitor that will not conduct DC current through it but will conduct AC current in a frequency-dependent manner. Therefore, AC frequencies can be used to probe the interior of nanoparticles with different levels of sensitivity; DC current mostly probes nanoparticle size and coating. Overall, the integration of AC sensing techniques into single-nanoparticle characterization protocols holds great promise for advancing the development and application of nanoparticle-based drug delivery systems.

*Mass photometry:* Mass photometry, commercialized by Refeyn (Oxford, UK), can be used to measure the molecular mass of single analytes such as proteins, nucleic acids, vesicles, and micelles. It can also be utilized for studying the biomolecular interactions such as protein–protein interactions or protein–DNA interactions [23]. Furthermore, it can be applied for characterizing AAV gene therapy vectors. Moreover, it has been implemented for SARS-CoV-2 research to study the oligomerization of the SARS-CoV-2 spike protein and the stoichiometry of its generation from the quantified light scattering from single-molecule analytes in solution, revealing the true molecular mass ranging from 30 kDa–5 MDa. For this technique, a beam of light is shone to a single-molecule analyte on the measurement surface, which results in a light scattering signal in which light is scattered by the molecule and light is reflected by the measurement surface, and this signal is proportional to the mass of a molecule. This method is advantageous because it is label-free, can be used in solutions, reveals molecular heterogeneity, and requires only a minimal amount of sample. 

*Real-time NTA:* Nanolyze (Mölndal, Sweden) has commercialized label-free image-based real-time NTA with a single-nanoparticle solution that can be used for size estimation and cargo loading of nanoparticles. The mRNA cargo loading heterogeneity of lipid nanoparticles is also a key quality indicator that determines the effectiveness of LNP-based therapy. This technology can also be used to study the interaction and structural changes in many biological and physical processes of liposome–liposome fusion, single-virus infection, nanoparticle aggregation, and protein–lipid interactions.

*Atomic Force Microscopy-Infrared Spectroscopy (AFM-IR):* Bruker Corporation (Billerica, MA, USA) has commercialized an AFM-IR tool for mapping and evaluating drug distribution within individual nanoparticles. This label-free, single-nanoparticle technique has been used to quantify the amount of cancer drug loaded within PLA nanoparticles with high precision [190]. Specifically, this technique quantified the drug mass fraction with a standard error of 5–6% and was able to quantify the chemical homogeneity at the nanoscale (10–20 nm) level. Importantly, the results of this analysis showed large drug load heterogeneities within nanoparticles, with only a small nanoparticle fraction being fully loaded with drug [190].

*Inductively Coupled Plasma Mass Spectrometry (ICP-MS):* Commercialized by PerkinElmer (Waltham, MA, USA) and Thermo Fisher Scientific (St. Bend, OR, USA), ICP-MS is particularly useful for bioassays involving metallic nanoparticles, offering high sensitivity and the ability to quantify drug load without complex signal amplification. Using this technique, one can obtain the nanoparticle size, size distribution, number density, mass concentration, mass agglomeration, and nanoparticle composition at low concentrations [191].

*Biolayer Interferometry-Surface Plasmon Resonance (BLI-SPR):* The commercialized Octet System from Sartorius company (Göttingen, Germany) is a label-free biosensor that can be used to characterize various samples such as antibodies, proteins, viruses, DNA/RNA, virus-like nanoparticles, etc., which can aid in studying the interaction between analytes, helping in the exploration of drug discovery and vaccine research. This system combines the technologies of Biolayer Interferometry (BLI) and Surface Plasmon Resonance (SPR), and this system provides real-time label-free measurements, which reveal information about the binding kinetics and affinity and can be used to determine the active concentration of an analyte by measuring the dissociation and dissociation rates. Furthermore, one of the advantages of this system is that it can provide fast quantification of up to 96 samples in 2 min, and there is no need for a pre-analysis purification step compared to current techniques such as HPLC and ELISA. The Octet system was able to distinguish the difference in binding kinetics of different monoclonal antibodies to treat non-Hodgkin’s lymphoma. 

## Figures and Tables

**Figure 1 bioengineering-12-00362-f001:**
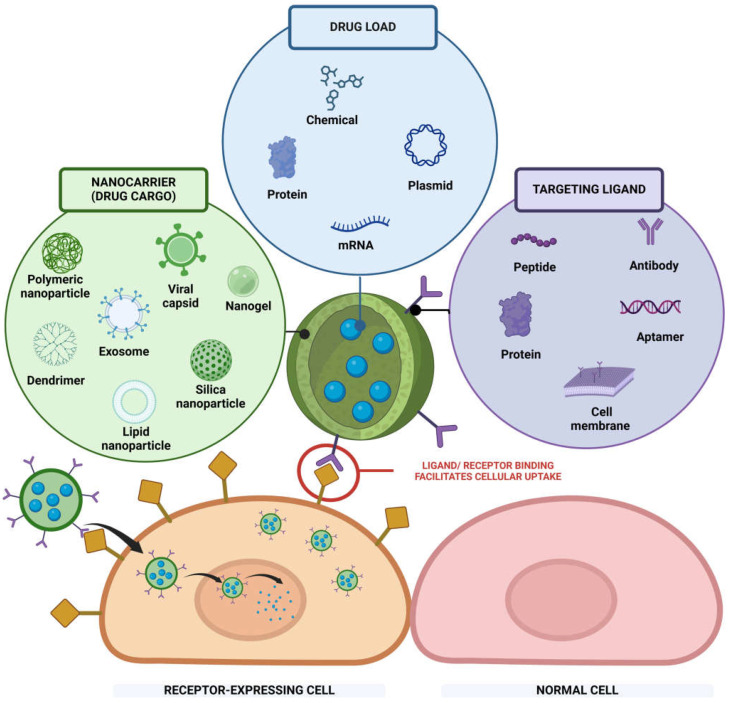
Conceptual schematic of nanocarriers for targeted therapeutic agent delivery.

**Table 1 bioengineering-12-00362-t001:** Methods for quantification of nanoparticle size distributions.

Category	Methods	Detection Range	Parameters That Can Be Measured	Advantages	Disadvantages	Sources
Bulk analysis	Dynamic light scattering (DLS)	0.3 nm–10 µm	Size, zeta potential, polydispersity	Can be conducted on a wide range of sample buffers, temperatures and concentrations. Non-invasive technique. Low amount of sample required [83]. Low peak resolution, can only resolve particle size at least by a factor of 3 [85].	Low resolution. Only for transparent sample preparation. Concentration needs to be optimized to produce reliable data [83].	[83,85,90]
	Small angle X-ray scattering	1–1000 nm [89]	Size, size distribution, shape, structure parameter, internal structure, crystallinity	Can determine various types of nanoparticles [90]. Little sample preparation time [90].	Low resolution, complexity in data interpretation, can be affected by solvent, sample preparation has to be dispersed.	[89,90]
	Field flow fractionation (FFF)	1 nm–hundreds of µm	Size, particle size distribution, molecular weight, shape, morphology, density, concentration	High separation efficiency, minimal sample requirement, real-time monitoring.	Not effective in differentiating small molecules.	[108]
	Size-exclusion chromatography (SEC)	10 kDa–1000 kDa	Size, size distribution	Preserve biological activity, fast, easy sample preparation.	Hard to differentiate populations of samples with similar sizes.	
	Centrifugal Sedimentation	0.01–40 µm	Size, size distribution, density, shape, concentration	High resolution, minimal sample preparation, rapid analysis, real-time monitoring	Causes damage to particles.	[91]
Single-molecule characterization	Tunable resistive pulse sensing	40 nm–11 µm (Izon) 40–20 µM [106]	Size, size distribution, zeta potential	More accurate. Can also be used to measure charge. Single-molecule analysis.	Clogging of analytes. Cannot measure large particles.	[106]
	Scanning Electron Microscopy (SEM)	Resolution: 3–20 nm	Size, polydispersity	High resolution.	Can only obtain surface information of nanoparticles [83].	[83]
	Transmission Electron Microscopy (TEM), Cryo-TEM	0.1 nm–10 µm	Size, polydispersity	Direct visualization. Can visualize the interior of the specimen. Cryo-TEM: can keep analytes in native form.	Expensive equipment. Complicated, time-consuming sample process. Only provides static and 2-dimensional information.	[103,109]
	Nanoparticle tracking analysis (NTA)	30–1000 nm [85] 30–600 nm [106]	Size, polydispersity, concentration	Enables sample visualization. Provides approximate concentration.	Requires optimization. More time-consuming than DLS.	[85,106]
	Atomic Force Microscopy (AFM)–Single molecule	0.5–50 nm	Size, deformability	Can see analytes’ topography. Allows 3D visualization [86].	Sample needs to be fixed. Deposition method will alter the size distribution [86].	[86]

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
