# Peer review of "Current and Near-Future Technologies to Quantify Nanoparticle Therapeutic Loading Efficiency and Surface Coating Efficiency with Targeted Moieties"

_bioengineering, 2025, doi:10.3390/bioengineering12040362_

Round 1
Reviewer 1 Report
Comments and Suggestions for Authors
The manuscript entitled “Current and near-future technologies to quantify nanoparticle therapeutic loading efficiency and surface coating efficiency with targeted moieties” reviews different instrumental methods that are currently available to assess size, shape, zeta potential, morphology, surface coating, drug loading and efficiency entrapment of nanoparticles as carriers for both drugs and genes. The authors compare advantages and disadvantages of bulk analyses with those of single-molecule and single-particle characterization assays. In addition, the authors suggest a list of specific techniques that could be used to characterize nanoparticles in the near future.
The manuscript provides the reader with useful information about the techniques used to characterize different types of nanoparticles, clearly highlighting the limits of each technique.
Although the manuscript is well organized, some inaccuracies should be corrected.
Line 56-60. The sentences “Nanoparticles are the new flatform in drug delivery, having many advantages such 56 as improved bioavailability, targeted delivery, and controlled release of therapeutic agents. These nanoscale carriers, which include liposomes, polymeric nanoparticles, and lipid nanoparticles (LNPs), are capable of transporting drugs directly to diseased tissues, minimizing side effects and enhancing treatment efficacy” repeat the same concepts expressed at line 29-33. Please, delete repetition.
Line 80. The term “solutions” is incorrect as nanoparticles are colloidal suspensions. Please, correct.
The meaning of the abbreviations should be explained the first time they are cited in the text. For instance, the meaning of AAV (line 104), Dex and Bos (line 165), EM (line 328), LNPs (line 349), MS and MALS (line 353), BCA (line 517), PCR (line 536), AC (line 732) should be explained. Please, revise all the manuscript explaining the meaning of the abbreviations.
Line 485. siRNA is repeated twice. Please, correct.
Line 529. ug should read µg. Pleases, correct.
In Table 2, ug should read µg. Please, correct. At line UV absorbance 205 nm, the application examples are missing and only a reference number is reported. Please, insert the application examples. The technique Mass spectrometry is listed in Table 2 but it is not discussed in the text. Please, discuss this technique in the text.
Line 625. The meaning of the sentence “The traditional for quantifying targeting ligands and may not be suitable for biological nano-particles such as liposomes which cannot endure the high separation methods” is unclear. Please, rephrase.
Line 630. The sentence “fluorescence-based method for directly quantifying antibodies bound onto gold nanoparticles” is a repetition of the previous sentence. Please, correct.
Line 631. The meaning of “KI/I2 etchant” is unclear. Please, explain.
Line 641. The meaning of the sentence “This paper compared between Bradford assay used with supernatant of the unbounded” is unclear. Please, rephrase.
English should be revised.
Author Response
We sincerely appreciate the reviewers’ time and thoughtful feedback. Their comments helped us improve the clarity and quality of our manuscript, and we’re grateful for their suggestions. Thank you for your valuable insights and for helping us strengthen our work.
Here are our response to some of your comments:
Comment 1: Line 56-60. The sentences “Nanoparticles are the new flatform in drug delivery, having many advantages such 56 as improved bioavailability, targeted delivery, and controlled release of therapeutic agents. These nanoscale carriers, which include liposomes, polymeric nanoparticles, and lipid nanoparticles (LNPs), are capable of transporting drugs directly to diseased tissues, minimizing side effects and enhancing treatment efficacy” repeat the same concepts expressed at line 29-33. Please, delete repetition.
Response: We agree with this comment, I deleted the second sentence.
Comment 2: Line 80. The term “solutions” is incorrect as nanoparticles are colloidal suspensions. Please, correct.
Response: we change the use of nanoparticles solution to nanoparticles suspension
Comment 3: The meaning of the abbreviations should be explained the first time they are cited in the text. For instance, the meaning of AAV (line 104), Dex and Bos (line 165), EM (line 328), LNPs (line 349), MS and MALS (line 353), BCA (line 517), PCR (line 536), AC (line 732) should be explained. Please, revise all the manuscript explaining the meaning of the abbreviations.
Response: I revised and spelled out all the addressed abbreviations above on the revised manuscript.
Comment 4: Line 485. siRNA is repeated twice. Please, correct.
Response: I deleted the second time using siRNA in the sentence.
Comment 5: Line 529. ug should read µg. Pleases, correct.
Response: I changed all the units from ug to µg on the revised manuscript
Comment 6: In Table 2, ug should read µg. Please, correct. At line UV absorbance 205 nm, the application examples are missing and only a reference number is reported. Please, insert the application examples. The technique Mass spectrometry is listed in Table 2 but it is not discussed in the text. Please, discuss this technique in the text.
Response: I changed all ug to be µg
Line 625. The meaning of the sentence “The traditional for quantifying targeting ligands and may not be suitable for biological nano-particles such as liposomes which cannot endure the high separation methods” is unclear. Please, rephrase.
Response: I want
Line 630. The sentence “fluorescence-based method for directly quantifying antibodies bound onto gold nanoparticles” is a repetition of the previous sentence. Please, correct.
Line 631. The meaning of “KI/I2 etchant” is unclear. Please, explain. I have explained
Line 641. The meaning of the sentence “This paper compared between Bradford assay used with supernatant of the unbounded” is unclear. Please, rephrase.
Reviewer 2 Report
Comments and Suggestions for Authors
In this revie article, the authors discussed about the challenges associated with nanomedicine for translation to bed side, and emerging technologies that have been using recently to overcome those challenges. This review article is scholarly written and would be of interest for researchers working in nanomedicine research field. So, this article could be accepted for publication after addressing following minor comment.
It is suggested to include summary images after each section and also some important images from discussed articles for quick understanding the concept.
Author Response
Thank you for your valuable feedback on our manuscript. We have carefully addressed the comments and attached the revised version, incorporating the suggested revisions. Please let us know if any further modifications are required.